# Adipokines Profile and Inflammation Biomarkers in Prepubertal Population with Obesity and Healthy Metabolic State

**DOI:** 10.3390/children9010042

**Published:** 2022-01-02

**Authors:** Lidia Cobos-Palacios, Mónica Muñoz-Úbeda, Cristina Gallardo-Escribano, María Isabel Ruiz-Moreno, Alberto Vilches-Pérez, Antonio Vargas-Candela, Isabel Leiva-Gea, Francisco J. Tinahones, Ricardo Gómez-Huelgas, María Rosa Bernal-López

**Affiliations:** 1Department of Internal Medicine, Instituto de Investigacion Biomedica de Málaga (IBIMA), Regional University Hospital of Málaga, University of Málaga, 29010 Málaga, Spain; cobospalacios@gmail.com (L.C.-P.); monicamubeda@gmail.com (M.M.-Ú.); carmencge@gmail.com (C.G.-E.); mruiz.salud@gmail.com (M.I.R.-M.); antonio.vargascandela@gmail.com (A.V.-C.); 2Department of Endocrinology and Nutrition, Instituto de Investigación Biomédica de Málaga (IBIMA), University Hospital Virgen de la Victoria, 29010 Málaga, Spain; alberto_v4@hotmail.com (A.V.-P.); fjtinahones@hotmail.com (F.J.T.); 3Department of Pediatric Endocrinology, Instituto de Investigacion Biomedica de Málaga (IBIMA), Regional University Hospital of Málaga, University of Málaga, 29010 Málaga, Spain; isabeleiva@hotmail.com; 4CIBER Fisiopatologia de la Obesidad y la Nutricion, Carlos III Health Institute, 28029 Madrid, Spain

**Keywords:** inflammation, lifestyle intervention, metabolically healthy obesity, prepubertal population

## Abstract

(1) Background and aims: Obesity and high body max index (BMI) have been linked to elevated levels of inflammation serum markers such as C-reactive protein (CRP), interleukin-6 (IL-6), tumor necrosis factor alpha (TNF-alpha), adiponectin, and resistin. It has been described that adipose tissue presents a high production and secretion of these diverse pro-inflammatory molecules, which may have local effects on the physiology of the fat cell and also systemic effects on other organs. Our aim was to evaluate the impact that lifestyle modifications, following a Mediterranean Diet (MedDiet) program and physical activity (PA) training, would have on inflammatory biomarkers in a metabolically healthy prepubertal population with obesity (MHOPp) from Malaga (Andalusia, Spain). (2) Methods: 144 MHOPp subjects (aged 5–9 years) were included in this study as they met ≤1 of the following criteria: waist circumference and blood pressure ≥ 90 percentile, triglycerides > 90 mg/dL, high-density lipoprotein cholesterol (HDL-c) < 40 mg/dL, or impaired fasting glucose (≥100 md/dL). Selected subjects followed a personalized intensive lifestyle modification. Anthropometric measurements, inflammation biomarkers, and adipokine profile were analyzed after 12 and 24 months of intervention. (3) Results: 144 MHOPp participants (75 boys—52% and 69 girls—48%; *p* = 0.62), who were 7.8 ± 1.4 years old and had a BMI 24.6 ± 3.3 kg/m^2^, were included in the study. After 24 months of MedDiet and daily PA, a significant decrease in body weight (−0.5 ± 0.2 SD units; *p* < 0.0001) and BMI (−0.7 ± 0.2 SD units; *p* < 0.0001) was observed in the total population with respect to baseline. Serum inflammatory biomarkers (IL-6, TNF-alpha, and CRP) after 24 months of intervention were significantly reduced. Adipokine profile (adiponectin and resistin) did not improve with the intervention, as adiponectin levels significantly decreased and resistin levels increased in all the population. Inflammatory biomarkers and adipokine profile had a significant correlation with anthropometric parameters, body composition, and physical activity. (4) Conclusions: After 24 months of lifestyle modification, our MHOPp reduced their Z-score of BMI, leading to an improvement of inflammatory biomarkers but inducing deterioration in the adipokine profile, which does not improve with MedDiet and physical activity intervention. An adequate education within the family about healthier habits is necessary to prevent and reduce an excessive increase in obesity in childhood.

## 1. Introduction

Obesity and overweight are the abnormal or the excessive fat accumulation, as defined by the World Health Organization (WHO). According to the WHO estimations, 41 million children under 5 years of age and more than 340 million children and adolescents (ages 5 to 19) were overweight or obese in 2016. The prevalence of overweight and obesity in children and adolescents has increased from 4% in 1975 to more than 18% in 2016 [1]. The main cause of obesity is a consumed/expended calories imbalance, although other factors can be involved. Globally, there is a trend towards consuming energy-dense foods, usually high in free sugars and fat, which is accompanied by a decrease in physical activity, mainly caused by changes in social and environmental factors [2,3]. The prevalence of obesity in childhood and youth population has become a public health challenge because of its increasing rate and the presence of several physical and psychosocial comorbidities. Moreover, the early morbi-mortality in this prepubertal population with obesity during the adult stage increases drastically, reducing life expectancy and producing nontransmissible illnesses such as diabetes and cardiovascular diseases [4]. In addition, obesity is a highly prevalent comorbidity in severe cases of COVID-19 [5] in childhood [6]. In severe acute respiratory syndrome coronavirus 2 infection, organic changes such as damage to immune, cardiovascular, and respiratory systems due to obesity may lead to alterations in the immune response, and, thus, to chronic inflammation, thromboembolism, and ventilatory assistance. Despite this, there is controversy in the literature about the anthropometric and biochemical study of children of less than 10 years old presenting obesity with one or more metabolic syndrome (MetS) criteria. A recent study based on the suggestion of the International Diabetes Federation about childhood and adolescent MetS with the incorporation of age-related developmental differences unique to the pediatric population has demonstrated that children less than ten years old cannot be diagnosed with a MetS due to the lack of data in this age group [7]. However, another study has demonstrated that following a Mediterranean Diet for 16 weeks, in children and adolescents presenting obesity with more than one MetS, improves their BMI, glucose, and lipid profiles [8].

Adipose tissue is a specialized organ that secretes different biomarkers that behave as paracrine and endocrine regulators [9]. These biomarkers include a variety of cytokines such as interleukin-6 (IL-6) [10], C-reactive protein (CRP), and tumor necrosis factor alpha (TNF-alpha), which decrease with dietary intervention and weight loss [11], and different adipokine profiles [12]. Adipokines (adiponectin and resistin) behave as autocrine/paracrine/endocrine mediators, regulating appetite, metabolism, and cardiovascular function [9]. Therefore, adipose tissue is responsible for a wide variety of inflammatory and metabolic interactions, which means that for maintaining a healthy phenotype it is essential that this tissue functions adequately. In addition, these chronic inflammatory markers (IL-6, CRP, and TNF-alpha, among others) are associated with several different diseases, including infection, cardiovascular disease [13,14], and type 2 diabetes mellitus (T2DM) [15,16].

Obesity is associated with chronic and cardiovascular diseases, but there are some people who despite presenting obesity (body mass index (BMI) ≥ 30 kg/m^2^) do not have metabolic complications. These subjects are known as metabolically healthy with obesity (MHO). Obesity and its associated diseases have been deeply studied. However, data from the MHO population are limited, particularly from the MHO prepubertal population (MHOPp) [17]. Although MHO does not necessary involve a lower risk of mortality, the implementation of a personalized dietary and physical activity (PA) treatment is important to avoid future diseases derived from obesity. Obesity is preventable and reversible. Energetic consumption modifications along with a regular PA can lead to weight loss. The Mediterranean Diet (MedDiet) is a dietary pattern characterized by high consumption of fruits, vegetables, legumes, and olive oil as the main source of fat intake, moderate-to-high fish intake, moderate consumption of wine (during meals and only for adult individuals), and low meat and poultry consumption [18]. MedDiet adherence is associated with lower risk of developing type 2 diabetes and cardiovascular diseases [19]. Youth health status is improved by the adherence to MedDiet, as several studies reported [20]. Childhood and adolescence are crucial periods for developing healthy habits. Regular PA in this age stage improves health in many ways. Compared to inactive children, active children tend to have stronger muscles and bones, less body fat, better academic results, and fewer symptoms of depression. Children and adolescents should do at least 1 h a day of moderate and vigorous PA [21].

The current study purpose was to explore the impact that 12 and 24 months of lifestyle modifications, by following a MedDiet program and daily PA training, would have on inflammatory biomarkers in MHOPp Spanish from Malaga (Andalusia).

## 2. Subjects and Methods

### 2.1. Study Design and Subjects

This study is a cross-sectional study of a MHOPp (4–9 years aged) that included participants of both sexes during a total period of 24 months. Inclusion criteria were boys (from 4 years to testicular volume < 3 mL) and girls (from 4 years to Tanner S2 breast bud elevation) with obesity (percentile ≥ 95) [22] and 1 or none of the following 4 cardio-metabolic disorders [23]: abdominal circumference and blood pressure ≥ 90 percentile, triglycerides > 90 mg/dL, HDL-c < 40 mg/dL, or impaired fasting glucose (≥100 mg/dL). Participants who had more than one MetS criteria, were diabetic, or had any metabolic illness were excluded from the study.

### 2.2. Procedure

The recruitment period was from November 2016 to May 2017 in different schools from the city of Malaga (Andalusia, Spain). These schools were located in the urban environment of the city and, in order not to bias the results, subjects with different socioeconomic statuses (low, medium, and high) and educational levels (medium–high/low educational level) were included in the study. Recruitment was carried out by performing preschool and elementary school classroom visits at different schools. Families of the possible selected participants were contacted to provide them with information about the study and were invited to participate in the program. The steps for the inclusion of children in the study were as follows: (a) acceptation of inclusion in the study by parents/legal representatives of the children; (b) verification that children met inclusion criteria; (c) interview with a nutritionist.

The parents/legal representatives and the selected subjects were summoned to the Department of Internal Medicine in the Regional University Hospital of Malaga so that they could be informed about the design and objectives of the study, as well as in order to request their written informed consent for the voluntary participation of their children.

All parents/legal representatives gave their informed consent to participate in this study, and protocols were approved by the institutional ethics committee (Code and Approbation: 07312015. Comité de Ética de la Investigación Provincial de Málaga, belonging to the Andalusian Health Service).

### 2.3. Visits

The subjects who participated in the study had different nurse and nutritionist visits; an initial visit, 12 months after, and at 24 months of intervention. Anthropometric measurements (weight, height, BMI, and waist circumference) and blood pressure were obtained by trained sanitary personnel. Food intake and PA questionnaires were completed in order to analyze the lifestyle intervention impact. A nonconsecutive, 3-day dietary record (two workdays and one weekend day), containing detailed information about food composition and cooking recipes over 72 h [24], and a food frequency questionnaire (number of times/day, number of days/week, number of days/14 days, number of days/months, rarely, or never) were completed in every visit [25]. Adherence to MedDiet was evaluated using a validated questionnaire of 14 items related to food consumption frequency, according to Trichopoulou A el al. [18]. As our participants belonged to a prepubertal population, the item related to red wine intake was omitted. Thus, very low adherence was considered to be <5 points, low adherence was 5–7 points, moderate adherence was 8–11 points, and high adherence, 12–13 points.

### 2.4. Assays

Anthropometric variables were measured by trained personnel according to the protocol. Weight and height were obtained through high-quality electronic calibrated scales and a wall-mounted stadiometer, respectively. BMI was calculated by dividing weight (kg) by height squared (m^2^). Obesity was defined as a BMI ≥ 25 kg/m^2^ for children. Waist circumference was measured halfway between the last rib and the iliac crest by using an anthropometric tape. Blood pressure was measured with a validated automated electronic sphygmomanometer (OMRON M7 (HEM-780-E, OMRON Healthcare Co. Ltd., Kyoto, Japan) after 5 min of rest while the participant was in a seated position. Blood samples were collected in order to know the inclusion criteria after an overnight fast, and biochemical analyses were performed in local laboratories using standard enzymatic methods.

Blood samples were processed to obtain blood serum, which was aliquoted and stored at −80 °C until its use. Inflammatory biomarkers and adipokines were measured in Research Laboratory of Instituto de Investigación Biomédica de Málaga (IBIMA). Serum adipokine and inflammatory biomarkers levels (IL-6 and TNF-alpha) were measured using an enzyme-linked immunosorbent assay (ELISA, R&D Systems, Inc., Minneapolis, MN, USA). For adiponectin levels, the minimum detectable concentration was 0.246 ng/mL. The intra- and inter-assay coefficients of variation were 3.5% and 6.5%, respectively. For resistin levels, the minimum detectable concentration was 0.026 ng/mL. The intra- and inter-assay coefficients of variation were 4.7% and 8.4%, respectively. For IL-6 levels, the minimum detectable concentration was 0.70 pg/mL. The intra- and inter-assay coefficients of variation were 2.6% and 4.5%, respectively. Lastly, for TNF-alpha levels, the minimum detectable concentration was 1.6 pg/mL. The intra- and inter-assay coefficients of variation were 4.7% and 5.8%, respectively. High-sensitivity CRP levels were measured using ELISA (DRG Instruments GmbH, Marburg, Germany). The minimum detectable concentration was 0.1 mg/mL. The intra- and inter-assay coefficients of variation were 4.4% and 3.3%, respectively.

### 2.5. Intervention

After having evaluated eating behaviors, families and children were guided to carry out lifestyle modifications to promote and incorporate a healthy diet. In this case, a caloric restriction was not implemented, due to the fact that the participants were growing children. The MedDiet recommended to the studied subjects included extra virgin olive oil and nuts consumption. The recommended caloric intake was 1500 kcal/day [26], distributed as follows: 30% of fat (5%–8% of saturated fatty acids, 15%–18% of monounsaturated fatty acids, 5%–8% of polyunsaturated fatty acids, and <300 mg of cholesterol/day), 50% of carbohydrates (<10% of simple sugars, 40% of complex sugars, and low glycemic index), and 20% of proteins [27,28]. A telephone number was given to the parents/legal representatives in case they had any questions on nutrition during the study.

A daily PA was also recommended following the internationally accepted PA guidelines [20]. In addition, subjects were provided with GENEActiv Actigraph GT3X+ (ActiGraph, Pensacola, FL, USA) accelerometers to collect several data. The accelerometer should be attached under the breast of the child with an elastic belt adjusted to ensure close contact with the body. Recordings were done every day for at least 7 days (weekdays and the weekend) to take the subjects PA and sleep hours, except during water-based activities.

Total body and regional body composition were estimated using dual-energy X-ray absorptiometry (DXA-Hologic, Discovery QDR Series, Bedford, MA, USA). Each subject was scanned distinguishing between bone and soft tissue. Edge detection and regional demarcations were done by computer algorithms (software version APEX 5.0, Hologic Horizon A, Bedford, MA, USA). The abdominal region was delineated by an upper horizontal border located at half of the distance between acromions and external end of iliac crests, a lower border determined by the external end of iliac crests, and the lateral borders extending to the edge of the abdominal soft tissue. The percentage of body fat, fat mass, lean mass, total mass, and total fat were indicated.

### 2.6. Statistical Analysis

Quantitative variables were expressed as mean ± standard deviation (SD), and qualitative variables were expressed as percentages. SD Z-scores for anthropometric parameters (weight, height, BMI, and systolic and diastolic blood pressures) were calculated according to Spanish Child Growth Standards [29]. Student *t*-test was used to compare quantitative variables, whereas the chi-square and Mann–Whitney tests were used for qualitative variables in order to contrast variables measured within each group at different time periods. Relationships between adipokine profile and inflammatory biomarkers (serum levels) and anthropometric and body composition parameters, adherence to the MedDiet, different physical activity intensities, and adipokine profile and inflammatory biomarkers (between them) were examined by means of Pearson correlation analysis.

To calculate the sample size (calculated using 180 SISA, Simple Interactive Statistical Analysis), we relied on other clinical studies that have shown the metabolic benefits of weight loss in the MHOPp [30], and assuming a 95% confidence level (*α* error of 0.5%), 80% statistical power, and a 5% loss rate, a sample of 110 MHO subjects is required. A recruitment of at least 130 participants was planned to take in account the possible nonparticipation or loss to follow-up. Statistical analysis was performed using SPSS for Windows, version 22.0 (IBM Corporation INC., Somers, NY, USA).

## 3. Results

The initial recruitment of population from 36 schools in Malaga included a total of 949 MHOPp, but only 172 participants came to the first visit at the Regional University Hospital of Malaga. Finally, after the participant rejection and the exclusion of participants due to not meeting the inclusion criteria, 144 MHOPp participants finally joined the study (Figure 1).

From these 144 participants, 75 (52%) were male and 69 (48%) female (*p* = 0.62), with an average age of 7.8 ± 1.4 years. All families presented a low–middle socioeconomic status (97.3%), with 42.3% of them showing a medium–high educational level and 57.7% of them a low one.

All results for different anthropometric parameters at the initial visit and after 12 and 24 months of lifestyle modification are shown in Table 1. Both male and female patients decreased their systolic and diastolic blood pressure after 24 months treatment, although this difference did not reach significance in any of the cases.

SD (Z)-scores for weight, height, BMI, and blood pressure (systolic and diastolic) for all participants are shown in Table 2. After 12 months of follow-up, a significant decrease in SDS weight (−0.6 ± 0.1 SD units in boys and −0.5 ± 0.2 SD units in girls; 0.5 ± 0.1 SD units in all population; *p* < 0.0001) and SDS BMI (−0.7 ± 0.1 SD units in boys and −0.4 ± 0.1 SD units in girls, *p* < 0.0001, respectively) was observed, and in the total population (−0.7 ± 0.2 SD units, *p* < 0.0001) with respect to baseline. Moreover, after 24 months of the intervention, children, both in the total population and according to gender, significantly decreased their SDS weight (−0.5 ± 0.1 SD units; *p* = 0.001) and their SDS BMI (−0.7 ± 0.1 SD units; *p* < 0.0001).

Regarding analytical parameters (Table 3), hsCRP serum levels also tended to decrease in all the population, although the difference was not statistically significant. IL-6 serum levels showed a significant reduction after 12 (1.1 ± 1.1 pg/mL, *p* < 0.0001) and 24 months of intervention (0.3 ± 0.3 pg/mL, *p* < 0.0001). TNF-alpha serum levels also decreased after the intervention (2.1 ± 1.6 pg/mL at baseline vs. 0.4 ± 0.3 pg/mL at 12 months vs. 1.1 ± 0.8 pg/mL at 24 months, *p* < 0.0001 at both periods of time). Concerning the adipokine profile, adiponectin serum levels decreased after intervention (9.8 ± 1.8 µg/mL at baseline vs. 6.6 ± 1.4 µg/mL at 12 months vs. 7.7 ± 4.0 µg/mL at 24 months, *p* < 0.0001 at both periods of time) and resistin serum levels increased in both sexes after 12 and 24 months of follow-up (3.7 ± 2.2 ng/mL at baseline vs. 5.2 ± 2.6 ng/mL at 12 months vs. 5.6 ± 2.4 ng/mL at 24 months, *p* < 0.0001 at both periods of time).

Changes in food intake and MedDiet adherence were also analyzed. Our participants decreased their energy intake after 12 months of follow-up (2180 ± 378 Kcal/day at baseline vs. 1874 ± 378 Kcal/day at 12 months, *p* < 0.0001) and after 24 months of intervention (2180 ± 378 Kcal/day at baseline vs. 1946 ± 1122 Kcal/day at 24 months, *p* < 0.177). They also reduced their total fat intake (101 ± 26 g/day at baseline vs. 85.3 ± 23.3 g/day at 12 months vs. 80.3 ± 20.2 g/day at 24 months, *p* < 0.0001 in both periods of time) and increased their fiber intake. Adherence to MedDiet improved (+3 points) after 24 months of intervention, especially due to the reduction in the consumption of red meat, sausages, hamburgers, and commercial pastries, and thanks to the increase in the daily consumption of vegetable and fruits. A decrease in butter, margarine, and cream consume was observed, with olive oil being the main source of fat used in cooking.

Multiple regression models adjusted for sex, baseline age, baseline BMI Z-score, and abdominal fat mass in baseline conditions for both adipokines were tested. For adiponectin and resistin serum levels, no associations were found. After 12 months of intervention, only delta SD BMI Z-score (*β* = −1.02, *p* = 0.005) showed association with adiponectin levels but for resistin levels no associations were found. After 24 months of intervention, for adiponectin serum levels, no associations were found and resistin levels only showed a significant association with SD (Z)-BMI (*β* = 1.14, *p* = 0.03).

The PA of our subjects also improved due to the intervention (Table 4). All of them increased their moderate daily PA after both 12 and 24 months of follow-up (28.0 ± 17.4 min/day at 12 months and 27.1 ± 16.1 min/day at 24 months, *p* < 0.0001 in both periods of time) and their vigorous daily PA (7.3 ± 6.4 min/day at 12 months and 8.0 ± 7.0 min/day at 24 months, *p* < 0.0001 in both periods of time).

On the other hand, participants increased significantly the fat mass, lean mass, and total mass. However, regarding the percentage of total fat, a significant decrement was observed in all the population, especially in girls, after 12 and 24 months of intervention (Table 5).

Correlation analyses between body composition and metabolic parameters are shown in Table 6. In baseline conditions, insulin levels and HOMA-IR presented a positive correlation with body composition. Total cholesterol and LDL-c correlated negatively with abdominal lean mass. Triglycerides levels presented a positive association with all abdominal components of body composition.

After 12 months of intervention, glycemic profile (glucose and insulin levels, HOMA-IR) was positively associated with abdominal body composition. It was replicated with triglycerides levels. After 24 months of intervention, only HDL-c presented a negative correlation with total fat.

Correlation analyses are showed in Table 7. In baseline conditions, only hsCRP presented a positive correlation with anthropometric parameters and body composition.

After 12 months of intervention, hsCRP was positively associated with WHI and abdominal fat mass. However, there were negative associations between IL-6 and the adherence to the MedDiet and PA, and between TNF-alpha and WHI, and adiponectin with anthropometric parameters, body composition, and PA.

After 24 months of intervention, there were positive associations between hsCRP and IL6 with WHI, and between TNF-alpha with SDS weight and resistin with all anthropometric parameters and body composition. On the other hand, adiponectin negatively correlated with weight, waist circumference, and WHI.

Positive correlations were observed between inflammatory biomarkers and adipokine profile, and IL-6 and TNF-alpha, and for resistin with hsCRP, IL-6, and TNF-alpha, in baseline conditions.

After 12 months of intervention, only IL-6 correlated positively with TNF-alpha, and after 24 months of intervention, only hsCRP presented a positive correlation with resistin levels and a negative correlation with adiponectin levels.

## 4. Discussion

The purpose of this study was to determine whether adherence to MedDiet and regular PA promote modifications in serum inflammatory biomarkers and adipokines levels after 12 and 24 months of lifestyle modification. The results obtained in our study show that, according to Z-score, the participants, as a whole and regardless of gender, lost weight and reduced their BMI, although obesity degree, measured by BMI, remained. Inflammatory biomarkers hsCRP, IL-6, and TNF-alpha significantly decreased after 12 and 24 months of intervention; however, adipokines profile did not improve despite the intervention. Nevertheless, these changes in BMI Z-score could be likely to translate into clinically important differences for subjects with obesity. BMI reduction can improve health status as Lycett K et al. showed [30]. Their findings show evidence that BMI measurements predict cardiometabolic risk in childhood.

The presence of obesity is conditioned by the socioeconomic level of the population. The families of our prepubertal population are characterized by presenting a medium–low socioeconomic level (97.3%) and an educational level with primary studies (57.7%). These conditions, together with the place of residence (urban environment, 100%), have favored the presence of obesity in our population; data in agreement with Papadimitriou A et al. [31,32] The greater availability of foods with high caloric content and the lack of physical activity due to new technologies (television and videogames), and the need to perform tasks in the shortest time as possible, as well as the convenience of the availability of a wide range of fast cooking and ultraprocessed foods, have favored this high prevalence of obesity in developed countries and specifically, in our environment.

It should be taken into account that over 35% of children with obesity under 10 years old, even under 5 years old, present one or more cardiovascular risk factor or MetS, such as arterial hypertension, dyslipidemia, disturbed glucose tolerance, or insulin resistance (IR) [33]. The MetS involves clinical alterations as cellular dysfunction, high fasting triglycerides and glucose, hypertension, low high-density lipoprotein cholesterol (HDL-col), systemic inflammation, and visceral obesity. There is still not an international consensus in the definition of childhood MetS. In addition, the study of only four components is accepted, such as abdominal obesity, arterial hypertension, dyslipidemia, and altered glucose metabolism, in the MetS concept. Therefore, MetS is diagnosed when abdominal obesity is accompanied by at least two of these other clinical alterations [34,35,36,37]. Although some studies about adipokines profile and inflammation biomarkers in children with obesity under 10 years of age with MetS are beginning to be designed, the MetS condition is not available yet for children because body proportions normally change during pubertal development and vary considerably depending of the different races and ethnic groups, and because the differences in waist-to-hip ratios are difficult to interpret in children [38,39]. For this reason, this population was not included as control in our study due to the possible lack of reliability in the data results. Recent studies have focused on the importance of fat mass distribution as a risk factor for cardiometabolic disease [40]. In our study, our prepubertal population showed an increment in fat mass after 24 m of intervention, but the percentage of total fat decreased in the whole population; data in concordance with Duran I et al. [41].

On the other hand, inflammatory biomarkers hsCRP, IL-6, and TNF-alpha are associated with obesity and other diseases, including inflammation, infection, heart attack risk, and cancer [42]. In our MHOPp, these inflammation biomarkers, especially IL-6 and TNF-alpha, reduced significantly after 12 and 24 months of lifestyle modifications. In the immune system, IL-6, a glycoprotein (secreted by macrophages, T cells, endothelial cells, and fibroblasts), promotes the differentiation and maturation of T and B lymphocytes, stimulates the production of immunoglobulin, and inhibits the secretion of pro-inflammatory cytokines such as TNF-alpha and IL-1. IL-6 has been proposed as a marker of MetS, since it is involved in many disorders related to excessive weight gain [43]. This leads to an accumulation of adipose tissue, which is associated with insulin resistance and cardiovascular disease. In our MHO population, the reduction of IL-6 levels after two different time periods (12 and 24 months) could be protecting this phenotype MHO from the development of metabolic disorders throughout life.

In addition, TNF-alpha is produced by macrophages/monocytes during acute inflammation and is responsible for several signaling events within cells, leading to necrosis or apoptosis. This protein is also important for resistance to infection and cancers [44]. In our participants, TNF-alpha decreased with respect to the basal levels after 24 months of follow-up, which indicates a significant inflammation reduction thanks to lifestyle modifications.

IL-6 is a marker of the MetS, but TNF-alpha is involved in insulin resistance and dyslipidemia. During physical activity, IL-6 is produced by muscle fibers and inhibits the production of TNF-alpha [45]. It is observed in our population with a feed-back between these two inflammatory biomarkers: when IL-6 increases its serum concentrations, TNF-alpha decreases, and vice versa. In addition, IL-6 enhances lipid turnover, stimulating lipolysis as well as fat oxidation. Regular physical activity induces suppression of TNF-alpha and thereby offers protection against TNF-alpha-induced insulin resistance. This effect was observed in our MHOPp after 12 months, however there was an increment of TNF-alpha levels when our participants decreased their light physical activity and increased their sedentarism. In addition, IL-6 is the first myokine that is produced and released by contracting skeletal muscle fibers, exerting its effects in other organs of the body [45]. It is involved in mediating the health-beneficial effects of exercise and it is involved in the protection against metabolic diseases such as T2DM or cardiovascular diseases.

On the other side, human obesity is also associated with a pro-inflammatory state, reflected in the elevation of others biomarkers such as CRP. Some studies provide evidence supporting a positive association between body composition and CRP [46]. Normal CRP values are <10 mg/L, values with less cardiovascular risk are <1 mg/L, while an intermediate risk is between 1–3 mg/L, and high risk > 3 mg/L. CRP levels can be controlled with a lifestyle modification through a balanced diet rich in antioxidants, daily physical exercise, and sleep [47]. In our study, hsCRP levels were reduced in the total MHOPp, but more significantly in girls than in boys, reducing their cardiovascular risks. It was shown that the concentration of CRP in the blood is higher in people (prepubertal and adults subjects) with overweight and obesity in relation to people with normal weight [48], possibly as a result of the production of IL-6 and TNF-alpha by adipose tissue.

Adipose tissue is not only involved in fat storage, it also has a crucial role as an endocrine organ, since it produces several bioactive substances, collectively called adipokines. Excess adipose tissue results in a deregulated production of these adipokines, which is related to the pathogenesis of various cardiovascular diseases. One of the most important adipokines is adiponectin, which is mostly expressed in subcutaneous adipose tissue. Its expression and blood concentration decrease as adiposity increases. This cytokine takes part in the regulation of glucose level and in the lipidic metabolism [49]. It is associated with insulin sensitivity and with anti-inflammatory, angiogenic, and vasodilator properties, which can have an influence on central nervous system disorders [50]. One effect of adiponectin is the reduction of TNF-alpha production in macrophages [51], counteracting its pro-inflammatory effects on the arterial wall and possibly protecting against arteriosclerosis, and it correlates negatively with BMI [52]. A lower adiponectin concentration is presented in populations with obesity, increasing their cardiovascular risk. Adiponectin inhibits liver neoglucogenesis and promotes fatty acid oxidation in skeletal muscle. In obesity, the pro-inflammatory effects of cytokines through intracellular signaling pathways involve the NF-kappa B and JNK systems. Our participants reduced adiponectin serum levels significantly after 12 months, regardless of their weight loss and their lifestyle changes. After 24 months of intervention, the serum levels increased, although they were lower than at baseline. These results are in concordance with those of Rondanelli M. et al. [53]. This may be because, despite their decrease in BMI Z-score, these subjects still presented obesity at the end of intervention [54]. On the other side, although in adults the inversely proportional relationship between circulating adiponectin levels and BMI is clear, in prepubertal population there are contradictory results regarding the correlation between adiponectin and BMI in this life period. Some studies report the existence of a negative correlation [55] or the absence of correlation [56] between both. Additionally, total adiponectin levels in children with obesity were reduced [57] and similar [56] to those of their lean peers.

Resistin, a cysteine-rich adipokine induced during adipogenesis, is proposed as a link between obesity and T2DM and may modulate numerous steps in the insulin-signaling pathway leading to insulin resistance and promote inflammatory processes [58]. Resistin is involved in human hepatic lipid and lipoprotein regulation by microsomal triglyceride transfer protein activation and induction of hepatocyte insulin resistance [59]. High resistin concentration increases the adiposity and accelerates the accumulation of low-density lipoprotein cholesterol (LDL-c) in arteries, increasing the risk of heart disease. Resistin can regulate pre-adipocyte differentiation; it can also behave as a pro-inflammatory molecule [60,61]. In our study, MHOPp subjects showed increased resistin serum levels after 12 and 24 months of MedDiet and PA. It is demonstrated that resistin upregulates the expression of TNF-alpha in PBMCs via the nuclear factor-*κ*B pathway.

Both adiponectin and resistin act on TNF-alpha, which may define a synergistic response between both adipokines. This pro-inflammatory biomarker decreased significantly after 12 months of intervention, but increased after 24 months of lifestyle modification, in line with what was observed in the expression levels of the adipokine profile. Although MHOPp subjects in our study remained with obesity after 24 months of intervention, according to Z-score, they decreased their weight and BMI, which translates into a significant reduction of inflammatory biomarkers. These data were corroborated by Gomez-Huelgas R et al. in MHO adult population [62] and imply a decrease in cardiovascular risk factor, as demonstrated by Reinehr T et al. [63]. The growth of children could explain the low decrease in BMI in the studied MHOPp, in concordance with the results obtained by Marti A et al. [64]. However, adipokine profile did not improve despite their weight loss and lifestyle changes.

## 5. Limitations

This study does not present random measurements or control subjects due to the lack of an international consensus in the definition of childhood MetS. Despite this, we believe that it is important that each participant compares his or her results at the end of follow-up to his or her conditions at baseline, rather than to others. The comparison of the data of the different parameters studied has been used to verify the efficacy of the 24-month lifestyle intervention in MHOPp.

## 6. Conclusions

Lifestyle intervention in prepubertal population based on consumption of MedDiet and daily PA training decreases pro-inflammatory biomarkers (CRP, IL-6, and TNF-alpha). In contrast, an increment of adiponectin levels was observed, although without reaching the basal concentration levels, after 24 months of lifestyle modification in MHOPp. An adequate education within the family about healthier habits to prevent and reduce an excessive increase in obesity in childhood is necessary.

## Figures and Tables

**Figure 1 children-09-00042-f001:**
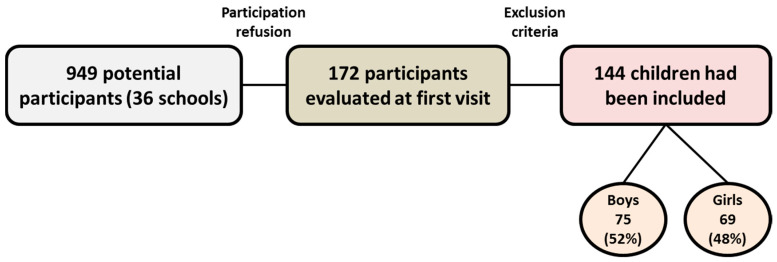
Screening and follow-up of studied metabolically healthy prepubescent population with obesity.

**Table 1 children-09-00042-t001:** Anthropometric parameters at baseline and after 12 and 24 months (m) of intervention in the total study population and by gender.

		Baseline (B)	12 Months (12 m)	24 Months (24 m)	*p*-Value (B vs. 12 m)	*p*-Value (B vs. 24 m)
Body weight (Kg) NV: 20.1–39.6	Boys	47.6 ± 10.0	50.7 ± 11.2	58.1 ± 12.7	<0.0001	0.002
	Girls	44.0 ± 10.3	49.9 ± 10.9	55.9 ± 11.4	<0.0001	0.16
	All	45.9 ± 10.3	50.3 ± 11.0	57.1 ± 12.2	<0.0001	<0.0001
Height (cm) NV: 114.4–144.2	Boys	137.8 ± 8.2	143.8 ± 8.5	150.4 ± 8.8	0.93	0.32
	Girls	133.3 ± 10.3	140.7 ± 8.9	148.7 ± 9.6	0.13	0.17
	All	135.7 ± 9.5	142.2 ± 8.9	149.7 ± 9.1	<0.0001	<0.0001
BMI (Kg/m^2^) NV:17.3–20.7	Boys	24.9 ± 3.6	24.3 ± 3.5	25.5 ± 3.9	<0.0001	<0.0001
	Girls	24.4 ± 3.1	24.9 ± 3.5	25.1 ± 2.9	<0.0001	0.003
	All	24.6 ± 3.3	24.6 ± 3.5	25. 3 ± 3.5	0.27	<0.0001
Waist circumference (cm) NV: 53.0–64.7	Boys	80.8 ± 9.6	82.4 ± 10.5	85.3 ± 11.5	0.003	<0.0001
	Girls	77.7 ± 7.4	80.0 ± 8.6	81.10 ± 8.6	0.06	<0.0001
	All	80.8 ± 9.6	82.4 ± 10.5	85.3 ± 11.5	<0.0001	<0.0001
SBP (mmHg) NV < p90	Boys	107.2 ± 10.9	109.4 ± 9.8	105.8 ± 14.5	0.86	0.03
	Girls	106.3 ± 13.7	110.3 ± 13.2	107.4 ± 11.9	0.08	0.72
	All	106.8 ± 12.3	109.9 ± 11.6	106.5 ± 13.4	0.01	0.91
DBP (mmHg) NV < p90	Boys	71.1 ± 11.0	69.6 ± 20.0	65.4 ± 26.9	0.28	0.32
	Girls	67.5 ± 9.5	72.3 ± 25.0	64.8 ± 26.3	0.38	0.68
	All	69.4 ± 10.4	71.0 ± 22.6	65.2 ± 26.5	0.27	0.60

NV: normal values for normal-weight subjects; BMI: body mass index; SBP: systolic blood pressure; DBP: diastolic blood pressure. Quantitative variables are expressed as mean ± SD. The paired *t*-test was used to compare quantitative variables in order to contrast variables measured in the entire prepubescent population at different times. *p*-value < 0.05 is significantly statistical.

**Table 2 children-09-00042-t002:** SD (Z)-scores at baseline and after 12 and 24 months (m) of intervention in the total study population and by gender.

		Baseline (B)	12 Months (12 m)	24 Months (24 m)	*p*-Value (B vs. 12 m)	*p*-Value (B vs. 24 m)
Body weight (Kg)	Boys	2.5 ± 1.4	1.9 ± 1.3	2.0 ± 1.3	<0.0001	0.002
	Girls	2.6 ± 1.6	2.1 ± 1.6	1.9 ± 1.3	<0.0001	0.16
	All	2.5 ± 1.5	2.0 ± 1.4	2.0 ± 1.3	<0.0001	0.001
Height (cm)	Boys	1.0 ± 0.8	0.9 ± 0.8	1.0 ± 0.9	0.93	0.32
	Girls	0.8 ± 1.3	0.6 ± 1.3	0.7 ± 1.3	0.13	0.17
	All	0.9 ± 1.1	0.8 ± 1.1	0.9 ± 1.1	0.32	0.30
BMI (Kg/m^2^)	Boys	2.6 ± 1.3	1.9 ± 1.3	1.9 ± 1.3	<0.0001	<0.0001
	Girls	2.6 ± 1.2	2.2 ± 1.3	1.9 ± 0.9	<0.0001	0.003
	All	2.6 ± 1.3	2.1 ± 1.3	1.9 ± 1.1	<0.0001	<0.0001
SBP (mmHg)	Boys	0.5 ± 1.0	0.5 ± 0.9	−0.02 ± 1.3	0.86	0.03
	Girls	0.6 ± 1.2	0.7 ± 1.2	0.3 ± 1.1	0.08	0.72
	All	0.5 ± 1.1	0.6 ± 1.1	0.1 ± 1.2	0.27	0.06
DBP (mmHg)	Boys	0.9 ± 0.9	0.7 ± 0.8	0.6 ± 1.0	0.28	0.32
	Girls	0.8 ± 1.0	0.9 ± 1.1	0.7 ± 1.2	0.38	0.68
	All	0.8 ± 1.0	0.8 ± 0.9	0.6 ± 1.1	0.99	0.33

BMI: body mass index; SBP: systolic blood pressure; DBP: diastolic blood pressure. Quantitative variables are expressed as mean ± SD. The paired *t*-test was used to compare quantitative variables in order to contrast variables measured in the entire prepubescent population at different times. *p*-value < 0.05 is significantly statistical.

**Table 3 children-09-00042-t003:** Analytical parameters at baseline and after 12 and 24 months (m) of intervention in the total study population and by gender.

		Baseline (B)	12 Months (12 m)	24 Months (24 m)	*p*-Value (B vs. 12 m)	*p*-Value (B vs. 24 m)
CRP (mg/L)	Boys	2.9 ± 2.4	2.6 ± 2.1	2.3 ± 2.2	0.74	0.33
	Girls	3.5 ± 3.1	2.6 ± 2.1	2.1 ± 2.1	0.09	0.05
	All	3.2 ± 2.8	2.6 ± 2.1	2.2 ± 2.1	0.14	0.05
IL-6 (pg/mL)	Boys	1.8 ± 1.6	1.0 ± 1.0	0.3 ± 0.3	<0.0001	<0.0001
	Girls	2.0 ± 1.6	1.2 ± 1.2	0.3 ± 0.3	<0.0001	<0.0001
	All	1.9 ± 1.6	1.1 ± 1.1	0.3 ± 0.3	<0.0001	<0.0001
TNF-alpha (pg/mL)	Boys	2.1 ± 1.5	0.4 ± 0.3	1.1 ± 0.7	<0.0001	0.004
	Girls	2.2 ± 1.8	0.5 ± 0.4	1.2 ± 1.0	<0.0001	0.004
	All	2.1 ± 1.6	0.4 ± 0.3	1.1 ± 0.8	<0.0001	<0.0001
Adiponectin (µg/mL)	Boys	9.8 ± 1.8	6.5 ± 1.7	7.6 ± 4.7	<0.0001	0.001
	Girls	9.9 ± 1.9	6.7 ± 1.1	7.8 ± 3.1	<0.0001	<0.0001
	All	9.8 ± 1.8	6.6 ± 1.4	7.7 ± 4.0	<0.0001	<0.0001
Resistin (ng/mL)	Boys	3.7 ± 2.4	5.0 ± 2.6	5.5 ± 2.6	0.04	0.005
	Girls	3.6 ± 2.1	5.5 ± 2.7	5.8 ± 2.1	0.001	0.003
	All	3.7 ± 2.2	5.2 ± 2.6	5.6 ± 2.4	<0.0010	<0.0001

CRP: C-reactive protein; IL-6: interleukin 6; TNF-alpha: tumor necrosis factor alpha. Quantitative variables are expressed as mean ± SD. The paired *t*-test was used to compare quantitative variables in order to contrast variables measured in the entire prepubescent population at different times. *p*-value < 0.05 is significantly statistical.

**Table 4 children-09-00042-t004:** Physical activity measured at baseline conditions, and after 12 months and 24 months of intervention in the total study population and by gender. (min/d: minutes/day). *p*-value < 0.05 is significantly statistical.

		Baseline (B)	12 Months (12 m)	24 Months (24 m)	*p*-Value (B vs. 12 m)	*p*-Value (B vs. 24 m)
Sedentarism (min/d)	Boys	397.0 ± 68.3	243.0 ± 78.4	338.1 ± 123.9	<0.0001	<0.0001
	Girls	397.1 ± 85.6	268.9 ± 92.1	399.8 ± 164.4	<0.0001	<0.0001
	All	397.0 ± 77.0	255.5 ± 85.2	367.6 ± 146.4	<0.0001	<0.0001
Physical activity						
Light (min/d)	Boys	663.6 ± 63.2	797.5 ± 91.9	363.9 ± 106.8	<0.0001	<0.0001
	Girls	671.4 ± 80.9	780.4 ± 92.1	308.6 ± 123.8	<0.0001	<0.0001
	All	667.4 ± 72.3	789.2 ± 92.0	337.7 ± 117.5	<0.0001	<0.0001
Moderate (min/d)	Boys	18.9 ± 18.0	31.7 ± 19.9	33.3 ± 16.5	0.01	0.02
	Girls	11.6 ± 11.5	24.0 ± 13.4	20.3 ± 12.7	<0.0001	0.001
	All	15.3 ± 15.5	28.0 ± 17.4	27.1 ± 16.1	<0.0001	0.001
Vigorous (min/d)	Boys	1.5 ± 3.9	7.8 ± 6.9	9.2 ± 8.1	<0.0001	<0.0001
	Girls	1.0 ± 1.8	6.7 ± 5.8	6.6 ± 5.6	<0.0001	<0.0001
	All	1.2 ± 3.1	7.3 ± 6.4	8.0 ± 7.0	<0.0001	<0.0001

**Table 5 children-09-00042-t005:** Body composition as measured by DXA at baseline, after 12 months, and after 24 months of intervention in the total study population and by gender (mean ± SD). *p*-value < 0.05 is significantly statistical.

		Baseline (B)	12 Months (12 m)	24 Months (24 m)	*p*-Value (B vs. 12 m)	*p*-Value (B vs. 24 m)
Fat Mass (Kg)	Boys	18.6 ± 5.4	19.0 ± 6.8	23.2 ± 8.4	0.2	<0.001
	Girls	18.5 ± 5.2	19.4 ± 6.0	21.4 ± 5.4	0.01	<0.001
	All	18.5 ± 5.3	19.2 ± 6.4	22.4 ± 7.2	0.01	<0.001
Lean Mass (kg)	Boys	28.1 ± 4.7	31.7 ± 4.9	34.8 ± 5.7	<0.001	<0.001
	Girls	25.8 ± 5.2	29.8 ± 5.9	33.3 ± 6.8	<0.001	<0.001
	All	27.0 ± 5.1	30.8 ± 5.5	34.1 ± 5.2	<0.001	<0.001
Total Mass (kg)	Boys	46.8 ± 9.7	50.6 ± 10.8	58.0 ± 7.1	<0.001	<0.001
	Girls	44.3 ± 9.8	49.2 ± 11.4	54.7 ± 6.1	<0.001	<0.001
	All	45.5 ± 9.8	49.9 ± 11.1	56.5 ± 6.2	<0.001	<0.001
Total Fat (%)	Boys	39.3 ± 4.1	36.8 ± 5.0	39.3 ± 5.7	<0.001	0.8
	Girls	41.4 ± 3.7	38.9 ± 4.0	39.0 ± 3.9	<0.001	0.001
	All	40.3 ± 4.0	37.8 ± 4.6	39.1 ± 4.9	<0.001	0.01

**Table 6 children-09-00042-t006:** Pearson correlation’s coefficient: body composition vs. metabolic parameters at baseline, and after 12 months and 24 months of intervention.

Baseline	Glucose		Insulin		HOMA-IR		Total Cholesterol		HDL-c		LDL-c		Triglycerides	
	r	*p*	r	*p*	r	*p*	r	*p*	r	*p*	r	*p*	r	*p*
Abdominal Fat Mass	0.08	0.35	0.55	<0.0001	0.53	<0.0001	−0.05	0.57	−0.09	0.29	−0.10	0.25	0.23	0.01
Abdominal Lean Mass	0.13	0.14	0.50	<0.0001	0.49	<0.0001	−0.19	0.03	−0.15	0.10	−0.23	0.01	0.22	0.01
Total Abdominal Mass	0.11	0.21	0.57	<0.0001	0.55	<0.0001	−0.11	0.20	−0.12	0.16	−0.17	0.06	0.24	0.01
Total Fat	0.01	0.87	0.39	<0.0001	0.37	<0.0001	0.06	0.54	−0.10	0.27	0.04	0.63	0.19	0.03
**12 Months**														
Abdominal Fat Mass	0.24	0.03	0.51	<0.0001	0.53	<0.0001	−0.01	0.93	−0.18	0.12	0.06	0.61	0.18	0.09
Abdominal Lean Mass	0.26	0.02	0.48	<0.0001	0.50	<0.0001	−0.08	0.48	−0.07	0.51	−0.12	0.28	0.26	0.02
Total Abdominal Mass	0.26	0.02	0.52	<0.0001	0.55	<0.0001	−0.04	0.74	−0.15	0.19	−0.02	0.89	0.22	0.04
Total Fat	0.14	0.22	0.38	0.001	0.39	<0.0001	0.07	0.56	−0.16	0.15	0.16	0.17	0.14	0.19
**24 Months**														
Abdominal Fat Mass	−0.08	0.59	0.22	0.14	0.15	0.31	−0.04	0.80	−0.20	0.15	0.05	0.72	0.04	0.79
Abdominal Lean Mass	0.01	0.96	0.22	0.14	0.18	0.22	−0.16	0.26	−0.03	0.85	−0.16	0.28	−0.10	0.51
Total Abdominal Mass	−0.05	0.74	0.24	0.11	0.18	0.24	−0.09	0.52	−0.14	0.31	−0.03	0.82	−0.02	0.92
Total Fat	−0.03	0.84	0.16	0.29	0.11	0.45	0.10	0.47	−0.29	0.04	0.22	0.12	0.20	0.16

**Table 7 children-09-00042-t007:** Pearson correlation’s coefficient: inflammatory biomarkers and adipokines vs. anthropometric parameters, Mediterranean Diet adherence and physical activity levels of intensity at baseline, after 12 months and after 24 months of intervention. m: months, BMI: body mass index, WC: waist circumference, WHI: waist/hip index, MedDiet: Mediterranean Diet.

Baseline	Weight		SDS Weight		BMI		SDS BMI		WC		WHI		Abdominal Fat Mass		Abdominal Lean Mass		Total Abdominal Mass		MedDiet Adherence	
	r	*p*	r	*p*	r	*p*	r	*p*	r	*p*	r	*p*	r	*p*	r	*p*	r	*p*	r	*p*
CRP	0.11	0.22	0.35	<0.0001	0.27	0.002	0.39	<0.0001	0.26	0.01	−0.01	0.91	0.30	0.001	0.11	0.23	0.24	0.01	−0.09	0.32
IL-6	0.09	0.29	0.05	0.58	0.08	0.35	0.05	0.58	0.07	0.41	−0.02	0.77	0.64	0.48	0.13	0.16	0.10	0.29	0.01	0.95
TNF-alpha	0.01	0.11	−0.34	0.69	0.05	0.53	−0.05	0.54	−0.01	0.91	−0.19	0.02	0.03	0.78	0.16	0.08	0.01	0.34	−0.27	0.75
Adiponectin	0.07	0.39	0.12	0.15	0.09	0.31	0.10	0.24	−0.03	0.71	0.06	0.45	0.15	0.10	0.12	0.18	0.14	0.11	0. 03	0.75
Resistin	0.13	0.15	0.07	0.39	0.11	0.22	0.07	0.42	0.13	0.13	−0.08	0.38	0.12	0.19	0.15	0.10	0.14	0.12	0.06	0.51
**12 Months**																				
CRP	0.16	0.15	0.11	0.30	0.19	0.07	0.16	0.14	0.20	0.07	0.29	0.01	0.24	0.03	0.08	0.45	0.19	0.08	−0.14	0.22
IL-6	0.10	0.34	0.04	0.68	0.05	0.62	0.03	0.81	0.05	0.66	−0.13	0.24	0.36	0.74	0.09	0.43	0.06	0.59	−0.28	0.01
TNF-alpha	0.06	0.56	−0.11	0.30	−0.04	0.71	−0.13	0.22	−0.03	0.75	−0.24	0.03	−0.06	0.59	0.03	0.76	−0.03	0.81	−0.18	0.09
Adiponectin	−0.20	0.07	−0.22	0.04	−0.21	0.05	−0.21	0.05	−0.28	0.01	−0.31	0.003	−0.23	0.03	−0.17	0.12	−0.22	0.04	0.07	0.54
Resistin	0.14	0.21	0.07	0.50	0.14	0.21	0.08	0.45	0.13	0.23	0.04	0.73	0.14	0.19	0.13	0.25	0.14	0.19	0.07	0.52
**24 Months**																				
CRP	0.06	0.62	0.14	0.24	0.17	0.16	0.20	0.10	0.20	0.10	0.36	0.002	−0.02	0.89	-0.12	0.40	−0.06	0.65	−0.15	0.20
IL-6	0.05	0.68	0.17	0.17	0.10	0.42	0.15	0.21	0.14	0.24	0.31	0.01	−0.004	0.98	−0.09	0.54	-0.04	0.78	−0.02	0.88
TNF-alpha	0.10	0.41	0.24	0.05	0.14	0.24	0.20	0.10	0.11	0.38	0.05	0.70	−0.01	0.92	0.09	0.53	0.03	0.83	−0.04	0.75
Adiponectin	−0.28	0.02	−0.20	0.09	−0.23	0.06	−0.17	0.16	−0.28	0.02	−0.23	0.05	−0.13	0.38	−0.12	0.40	−0.13	0.35	0.06	0.60
Resistin	0.46	<0.0001	0.45	<0.0001	0.49	<0.0001	0.46	<0.0001	0.37	0.001	0.23	0.05	0.43	0.002	0.28	0.05	0.40	0.004	−0.09	0.46
	**Physical Activity**																			
**Baseline**	**Light**		**Moderate**		**Vigorous**		**Moderate to** **Vigorous**		**CRP**		**IL6**				**TNF-alpha**		**Adiponectin**		**Resistin**	
	**r**	** *p* **	**r**	** *p* **	**r**	** *p* **	**r**	** *p* **	**r**	** *p* **	**r**		** *p* **		**r**	** *p* **	**r**	** *p* **	**r**	** *p* **
CRP	0.08	0.40	−0.06	0.54	−0.05	0.60	−0.06	0.54	-	-	−0.04		0.66		−0.09	0.32	0.10	0.24	0.28	0.001
IL-6	0.12	0.20	−0.06	0.51	−0.12	0.19	−0.07	0.43	−0.38	0.66	-		-		0.35	<0.0001	0.01	0.92	0.17	0.05
TNF-alpha	0.06	0.55	−0.04	0.68	−0.07	0.44	−0.04	0.63	−0.09	0.32	0.35		<0.0001		-	-	0.01	0.88	0.37	<0.0001
Adiponectin	0.05	0.60	0.05	0.57	0.11	0.24	0.06	0.49	0.10	0.24	0.01		0.92		0.01	0.88	-	-	−0.08	0.38
Resistin	0.12	0.21	−0.05	0.58	−0.08	0.37	−0.06	0.52	0.28	0.001	0.17		0.05		0.37	<0.0001	−0.08	0.38	-	-
**12 Months**																				
CRP	0.002	0.99	−0.19	0.10	0.001	0.99	−0.15	0.20	-	-	0.15		0.89		−0.14	0.22	−0.11	0.31	0.17	0.13
IL-6	−0.26	0.02	−0.24	0.03	−0.16	0.16	−0.24	0.03	0.01	0.89	-		-		0.54	<0.0001	−0.04	0.70	−0.17	0.88
TNF-alpha	−0.16	0.15	−0.07	0.54	−0.04	0.73	−0.07	0.56	−0.14	0.22	0.55		<0.0001		-	-	0.01	0.90	0.07	0.50
Adiponectin	0.21	0.06	0.29	0.01	0.17	0.13	0.28	0.01	−0.11	0.31	−0.04		0.70		0.01	0.90	-	-	0.10	0.38
Resistin	0.03	0.79	0.44	0.67	0.10	0.38	0.06	0.57	0.17	0.13	−0.17		0.88		0.07	0.50	0.10	0.38	-	-
**24 Months**																				
CRP	0.04	0.76	0.02	0.90	−0.09	0.52	0.06	0.52	-	-	0.17		0.15		0.02	0.84	−0.29	0.01	0.24	0.04
IL-6	−0.08	0.53	−0.01	0.93	−0.10	0.45	−0.10	0.43	0.17	0.15	-				0.13	0.28	−0.02	0.86	0.11	0.35
TNF-alpha	−0.02	0.88	−0.02	0.88	−0.06	0.67	−0.08	0.53	0.02	0.84	0.13		0.27			-	0.05	0.71	−0.20	0.87
Adiponectin	0.10	0.44	0.18	0.17	0.21	0.10	0.06	0.49	−0.29	0.01	−0.02		0.86		0.05	0.71	-	-	−0.10	0.42
Resistin	0.06	0.61	−0.10	0.45	0.03	0.82	−0.06	0.52	0.24	0.04	0.11		0.35		−0.20	0.87	−0.10	0.42	-	-

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
