# Peer review of "Adipokines Profile and Inflammation Biomarkers in Prepubertal Population with Obesity and Healthy Metabolic State"

_children, 2022, doi:10.3390/children9010042_

Round 1

Reviewer 1 Report

In this study including 144 metabolically healthy obese prepubertal children (MHOPp) from Malaga (Andalusia, Spain), the authors evaluated the impact of lifestyle modifications, following a Mediterranean Diet (MedDiet) program and physical activity (PA) training, on body mass and composition and inflammatory biomarkers. After 24 months of lifestyle modification, the MHOPp reduced their (z)-score of BMI, leading to an improvement of inflammatory biomarkers but induced deterioration in the adipokine profile, which did not improve with MedDiet and physical activity intervention.

Comments:

  1. When the authors state that obesity is ‘… reducing life expectancy and producing non-transmissible illnesses such as diabetes and cardiovascular diseases.’, the authors need to discuss that obesity is now also considered an important risk factor for severe COVID-19 (Nat Rev Endocrinol. 2021 Mar;17(3):135-149).
  2. What cut-off value was used to determine impaired fasting glycemia?
  3. The age range of the children was 4-9 years. All children had the same recommended caloric intake, 1500 kcal/day. This is surprising, as caloric intake should be lower in girls, compared to boys, and only about 1200 (girls) and 1400 (boys), e.g. as recommended to 4-8 year old children by the US Heart Association guidelines.
  4. How come that the baseline caloric intake of these young children was that high: 2180±378 Kcal/day?
  5. What were the characteristics and socio-economical background of the 777 children who did not show up to the first visit and how did they compare to the 172 participants who came to the first visit?
  6. How was adherence to the Med diet estimated?
  7. How were abdominal lean mass, abdominal fat mass and total abdominal mass measured?
  8. What were the changes of these parameters during the intervention?
  9. Can the authors prove that ‘all subjects had a weight loss and reduced their BMI’?
  10. The authors need to discuss that it would have been important to also investigate changes of lower body fat mass, which is now established as an independent and cardiometabolically protective fat depot (Lancet Diabetes Endocrinol. 2020 Jul;8(7):616-627), and relate these changes to changes in metabolic parameters.
  11. The authors extensively discuss possible effects of the adipokines and cytokines studied for metabolic health. However, they also need to discuss that fatty liver, which is a main characteristic of insulin resistance and MUHO (Lancet Diabetes Endocrinol. 2019 Apr;7(4):313-324) and predicts less improvement of MUHO during lifestyle intervention in adults (Lancet Diabetes Endocrinol. 2018 Mar;6(3):249-258), may also be strongly predictive of the changes in the metabolic parameters in the present study.

Author Response

Dear reviewer,

We thank you very much for giving us the opportunity to revise our manuscript. We have carefully considered your comments and we agree with most of them. Each comment has been addressed and we have modified the manuscript accordingly.

We sincerely hope that the current version of the manuscript will be acceptable for publication in your journal.

All changes are shown in red so that they may be easily seen.

Dr. Maria Rosa Bernal-Lopez

Reviewer 2 Report

It is an interesting study that emphasizes once again the importance of prevention, because not all pathological changes that occur can be fully corrected later.

I recommend you to add a list of abbreviation. Also, I recommend you to add the limits of the study before the conclusions.

Congratulation for your work! 

Author Response

(The authors gave the same response as above.)

Reviewer 3 Report

The study is performed in a population of 144 prepubertal children aged 5 to 9 years, defined as metabolically healthy and followed for two years. The definition is based on the fact that these children do not have more than one criterion included in the definition of metabolic syndrome. As also pointed out by the authors, it should be mentioned that there is no accepted definition of metabolic syndrome in this age group. Therefore, in pre-pubertal children, it would be better to speak simply of cardiovascular risk factors. The study focuses attention on changes in biomarkers of inflammation and two adipokines: adiponectin and resistin.

The study provides a lot of data, however in my opinion there are several points to clarify:

- The main problem is that there is a lack of multivariable analysis of the results, which prevents them from being properly evaluated. The values of the adipokines considered are certainly influenced by changes in age, weight, waist circumference and pubertal stage. Please make a multivariable statistical model that takes into account these confounding factors.

- What type of intervention was offered to study participants? Were follow-up visits scheduled? If so, with what frequency and intervention modalities?

- Was there any drop out or did all 144 children complete the study and complete the planned follow-up?

- During the two years of follow-up did any children begin pubertal development (Tanner classification >1)?

- During the 2-year follow-up did any children develop other cardiovascular risk factors, eg, blood pressure > 90th percentile?

- At baseline, LDL cholesterol, HDL cholesterol, triglycerides, and blood glucose were measured. At follow-up these parameters were no longer measured? For what reason? If these data are there, please show them.

- Were uric acid and insulin not measured?

- How is obesity defined? BMI > 95th percentile is mentioned (line 133). What reference nomograms were used? WHO, IOTF, local tables? On line 178 it says that "Obesity was defined as a BMI ≥ 25 kg/m2 for children" what does this statement mean? Is it possibly referring to the IOTF classification (Cole BMJ 2000)?

- How were abdominal circumference and blood pressure classified? When talking about values > 90th percentile what are the reference tables used?

- Blood pressure is measured only once, whereas guidelines suggest taking three measurements and eliminating the first, which is often higher than the others.

- Table 1 defines normal blood pressure values as 135/80 mmHg. These values are definitely pathological for children in the age range of the study population.

- In Table 5 there are columns referring to abdominal mass (total, fat, and lean). There is no mention of this measurement in the methods or text. Please explain.

- In the part of the methods related to the intervention it is stated: "a caloric restriction was not implemented, due to the fact that the participants were growing children". Also in growing children should not be administered a caloric excess, but should be prescribed a normocaloric diet. Subsequently it is written "The MedDiet recommended to the studied subjects included extra virgin olive oil and nuts comsumption. The recommended caloric intake was 1500 kcal/day" (lines 209-211)". Was the daily caloric intake calculated or not? Were participants also given quantitative indications in the diet or only qualitative ones (Mediterranean diet)? Were all subjects prescribed a diet of 1500 kcal/day? This could be an error, nutritionally speaking. In a child with a height on the 5th percentile the daily caloric requirement is about 1160 Kcal, while in a child with a height on the 95th percentile the daily caloric requirement is about 1730 kcal (basal metabolism according to Schofield (1985) x 1.3 basal metabolism factor for a predominantly sedentary activity). If all children were prescribed a diet of 1500 kcal some were instructed to consume too many calories and others to consume too few. In the text it is then later stated that the average caloric intake went from 2180 to 1874 at 12 months and to 1946 at 24 months. This further confused me. Please clarify.

- I would ask the authors to show data at baseline, 12 months, and 24 months to get a better idea of the trend of the different parameters during follow-up.

- The data for biomarkers of inflammation are interesting. However, the two adipokines have a trend opposite to the expected. In fact, not only do they not improve, but they seem to worsen. The explanation that the authors give for this phenomenon, namely that the children, although improved in terms of BMI, remained obese, does not seem satisfactory to me.

In conclusion, the study has many weaknesses. The authors focus on changes in inflammatory biomarkers and adipokines, while in my opinion it would be very interesting to evaluate the effectiveness of the intervention over the two-year follow-up also on other available cardio-metabolic risk parameters.

Author Response

(The authors gave the same response as above.)

Round 2

Reviewer 1 Report

The authors have satisfactorily addressed the critical points.

Author Response

Thank you very much

Reviewer 3 Report

I think the authors have only partially responded to my requests.

In detail:

Response: The reviewer is right. The requested analyses were made but, in our prepubertal population, these confounding factors (age, weight, and waist circumference) did not affect our results (inflammatory biomarkers and adipokines levels). On other side, the participants were in prepubertal stage during the complete study, measured by testosterone in boys and LH and 17 beta-estradiol levels for girls. We measured these hormones when all participants reached 8 years old (boys and girls). 

I'm sorry, but it is not enough for the authors to state that they have done a multivariate analysis. The authors should, please, show the results of the model. With Pearson's test, we should expect a negative correlation between BMI (BMI z-score) and WC (WCI/abdominal fat) and adiponectin, and, conversely, there should be a positive correlation between BMI (BMI z-score) and WC (WCI/abdominal fat) and resistin. This is actually true at 24 months, but not at baseline. The finding is odd. Only a multivariate analysis of the role of changes in BMI and WC from baseline and follow-up could provide explanations. Pearson's correlation's coefficient is a univariate analysis. Without a multivariate analysis, the statement of the conclusions of the abstract is not justified. "After 24 months of lifestyle modification, our MHOPp reduced their (z)-score of BMI, leading to an improvement of inflammatory biomarkers but inducing deterioration in the adipokine profile." I would suggest running the following statistical models.

1) a multiple regression model adjusted for sex, baseline age, baseline BMI-z-score and WtHr (or, better, abdominal fat mass) that has as outcome the value of the two cytokines at baseline.

This first analysis would allow to understand which factors are independently associated with the two cytokines studied. Another model should then be run to understand which factors are associated with changes in the value of the two cytokines at follow-up.

2) A multiple regression model adjusted for sex, follow-up age, baseline BMI-z-score and delta BMI z-score (baseline-follow-up), baseline WtHr and delta (baseline-follow-up WtHr) (or, better, baseline abdominal fat mass and delta abdominal fat mass (baseline-follow-up)) that has as outcome the value of the two cytokines at follow-up.

Response: No, all the population continued in prepubertal stage, and it was checked by hormones levels (testosterone, LH, and 17 beta-estradiol).

The age range of the sample studied was 4-9 years, and follow-up lasted two years. So some children at the end of the follow-up were 11 years old. It seems odd to me that none of the study participants, particularly if they were female, began pubertal development. Are the authors confident in this finding? Incidentally, the change in the relationship between anthropometric variables and the two adipokines would suggest that some new factor has intervened. One factor is certainly increasing age, but another could be the onset of puberty. In fact, an association between obesity and early puberty has been demonstrated, particularly in girls (Pediatrics 2010 September 126(3): e583-e590. doi:10.1542/peds.2009-3079) The link between childhood obesity and early puberty would seem to be due to hyperinsulinemia, which is very frequent in obese children.

Author Response

Journal: Children

Manuscript ID: children-1440915

Title: Adipokines profile and inflammation biomarkers in MHO prepubertal population Authors

Authors: Lidia Cobos-Palacios, Monica Muñoz-Ubeda, Cristina Gallardo-Escribano, M Isabel Ruiz-Moreno, Alberto Vilches-Perez, Antonio Vargas-Candela, Isabel Leiva Gea, Francisco J Tinahones, Ricardo Gomez-Huelgas, M Rosa Bernal-López *

Dear Reviewer

We thank you very much for giving us the opportunity to revise our manuscript. We have carefully considered your comments. Each comment has been addressed. 

We sincerely hope that the current version of the manuscript will be acceptable for publication in Children.

Dr. Bernal-Lopez
